# Invasive Lobular Carcinoma: A Review of Imaging Modalities with Special Focus on Pathology Concordance

**DOI:** 10.3390/healthcare11050746

**Published:** 2023-03-03

**Authors:** Alicia M Pereslucha, Danielle M Wenger, Michael F Morris, Zeynep Bostanci Aydi

**Affiliations:** 1Department of Surgery, University of Arizona College of Medicine-Phoenix, Phoenix, AZ 85006, USA; 2College of Medicine-Phoenix, University of Arizona, Phoenix, AZ 85004, USA; 3Division of Diagnostic Imaging, Banner MD Anderson Cancer Center, Phoenix, AZ 85006, USA; 4Department of Radiology, Banner University Medical Center-Phoenix, Phoenix, AZ 85006, USA; 5Department of Surgical Oncology, Banner MD Anderson Cancer Center, Phoenix, AZ 85006, USA

**Keywords:** invasive lobular cancer, breast, imaging, MRI, concordance, contrast-enhanced mammogram, radiomics

## Abstract

Invasive lobular cancer (ILC) is the second most common type of breast cancer. It is characterized by a unique growth pattern making it difficult to detect on conventional breast imaging. ILC can be multicentric, multifocal, and bilateral, with a high likelihood of incomplete excision after breast-conserving surgery. We reviewed the conventional as well as newly emerging imaging modalities for detecting and determining the extent of ILC- and compared the main advantages of MRI vs. contrast-enhanced mammogram (CEM). Our review of the literature finds that MRI and CEM clearly surpass conventional breast imaging in terms of sensitivity, specificity, ipsilateral and contralateral cancer detection, concordance, and estimation of tumor size for ILC. Both MRI and CEM have each been shown to enhance surgical outcomes in patients with newly diagnosed ILC that had one of these imaging modalities added to their preoperative workup.

## 1. Introduction

Invasive lobular cancer (ILC) is the second most common type of breast cancer after invasive ductal cancer (IDC) with an estimated 28,785 new cases of ILC in the United States in 2022 [1]. Even though ILC comprises only about 10% of new invasive breast cancer diagnoses, its incidence is high and similar to that of ovarian cancer (estimated 19,880 new cases in 2022) and twice that of cervical cancer (estimated 14,100 new cases in 2022) [1,2]. ILC is associated with older age, more advanced stage at presentation, and larger size and nodal positivity in comparison to IDC [3]. The majority of ILCs are Luminal A intrinsic subtype as evidenced by high estrogen receptor (ER) and progesterone receptor (PR) expression and low HER2 amplification. Histologic subtypes of ILC include classical, pleomorphic, signet ring, solid, alveolar, tubulolobular, and histiocytoid carcinomas [4]. Pleomorphic variants harbor higher rates of HER2 amplification (35%) and triple negative cancers (13%) [5].

ILC is characterized by a unique growth pattern where the stromal invasion causes little disturbance of the normal architecture. Cancer cells grow in single files with minimal desmoplastic stromal response. ILC is characterized by loss of E-cadherin. E-cadherin is a calcium dependent transmembrane protein that maintains tissue integrity by maintaining cell-to-cell adhesion and preventing invasion [6]. It is coded by the CDH-1 gene located on 16q22.1 [7]. Loss of E-cadherin results in loss of alpha-, beta-, and gamma-catenins, while p120-catenin is upregulated and accumulates in the cytoplasm and plays an important role in invasion. Loss of E-cadherin and cytoplasmic accumulation of p120 are present in >90% of ILCs [8].

ILC has been historically difficult to detect on conventional breast imaging as it is often isodense or isoechoic to adjacent normal breast parenchyma and does not present as a palpable mass [9], also making it difficult to discern on clinical examination. Incomplete excision of ILC is common after breast conservation surgery, ranging from 12% to 60% [10,11,12], leading to re-excisions or even mastectomy. Positive margins can also be present after mastectomy with extensive ILC [13]. Moreover, ILC can be multicentric/multifocal and bilateral, making preoperative imaging workup critical to achieve complete resection [14]. In this review, we aim to summarize the performance of different imaging modalities in detection of ILC and to determine the extent of tumor burden within the breast.

## 2. Methods

A comprehensive literature search of the PubMed database was conducted using the following keywords: ‘invasive lobular carcinoma’, ‘pathology’, ‘imaging’, ‘concordance’, and each of ‘mammogram’, ‘ultrasonography’, ‘tomosynthesis’, ‘MRI’, and ‘contrast-enhanced mammogram’. A supplemental search was conducted to include more recent modalities including ‘computed tomography’, ‘positron emission mammography’, ‘positron emission tomography’, ‘molecular breast imaging’, and ‘artificial intelligence’ to include more recent modalities. Articles published between 1995 and 2022 including books, meta-analyses, clinical trials, randomized control trials, and reviews, as well as relevant publications that were cited in the selected articles were considered; case reports were excluded. After database research was concluded, articles were manually selected by authors after reviewing the abstract and/or manuscript with regard to relevance. Selected articles in English from the database search were included in this review.

## 3. Results

### 3.1. Mammography and Digital Breast Tomosynthesis

Mammography (MMG) is the most commonly performed breast imaging modality and is the only screening test proven to reduce breast cancer mortality in randomized controlled trials [15]. Identification of ILC on mammography has proven to be a diagnostic challenge, and ILC carries a false negative rate higher than for other invasive cancers [16]. Because of the permeative growth pattern of ILC, up to 30% are mammographically occult and the pathologic size of ILC tumors can be underestimated in up to 70% of mammograms [17]. Mammographic sensitivity for ILC ranges from 34% to 92%, which trends lower than the mammographic sensitivity for IDC ranging from 63 to 98% [16,17,18,19,20]. The sensitivity of digital mammography (DM) is inversely related to breast density [21], which is exacerbated by the subtle findings of ILC. A recent literature review found that the sensitivity of DM across all densities is estimated to be around 34–83% for ILC but only around 8–11% for ILC in dense breasts [17].

The features of ILC on mammography can be subtle and variable. ILC most commonly presents as a spiculated mass (on DM) or architectural distortion (on mammography with tomosynthesis), followed by an asymmetric density or calcifications [22,23]. Less than 1% of ILC presents as well-circumscribed masses with regular borders [16,20]. Utilization of digital breast tomosynthesis (DBT) may improve the sensitivity and area under the curve for detection of ILC compared to digital mammography [24]. In a large multicenter trial, compared to DM alone, DBT identified more invasive breast cancers, and the detection rate for ILC increased from 0.27 to 0.55 per 1000 [25]. Similarly, Krammer et al. reported that DBT detected more ILCs in dense breasts than DM alone with increased sensitivity from 38.9% to 83.3% [26]. DBT significantly increased the conspicuity of ILC lesions (*p* = 0.002) compared to DM whereas the difference in conspicuity for IDC lesions was not significant (*p* = 0.2) [23].

While DBT can increase the conspicuity of ILC lesions, it fails to accurately assess tumor size of these lesions. Mann et al. noted that, in patients where ILC tumor was detected by mammography, more than half had an underestimation of tumor size by about 1 cm. In one review of imaging modalities for diagnostic evaluation for patients with suspected or proven ILC, it was referenced that large ILCs (measuring > 3 cm) were found to be particularly difficult to accurately measure on mammography [16]. Likewise, DM has been found to perform poorly for detecting multifocal, multicentric, and bilateral disease [16,17,26]. Overall, DBT can improve the detection of ILC when compared to DM alone, though imaging findings are subtle. It is not an adequate alternative to more sophisticated alternatives such as breast magnetic resonance imaging (MRI) and contrast-enhanced mammography for assessment and pretreatment workup of ILC. Given its superior performance compared to DM alone, it may be an alternative for imaging-guided biopsies of lesions detected by MRI [17].

### 3.2. Ultrasonography

Whole-breast ultrasonography (US) is not widely used for breast cancer screening given the higher rate of false positives relative to mammography [27]. However, diagnostic breast US is the imaging modality of choice for further characterization of a palpable abnormality or suspicious mass, distortion, or asymmetry visualized on mammography [28].

US sensitivity for ILC tumor detection is lower than other imaging modalities with sensitivity rates ranging from 68% to 92% [29,30,31]. Unlike mammography, US sensitivity is not impacted by breast density [32].

US has also been found to underestimate ILC tumor size in 18–53% of cases [29,33,34]. US-measured ILC tumor size and volume has also been found to correlate with final pathology to a lesser degree than other imaging modalities [35,36,37]. Vijayaraghavan et al. reported that for ILC tumors there is a median underestimation of tumor size by 3.5 mm (average 27.2%) and a median underestimation of tumor volume by 0.29 cm^3^. The extent of ILC tumor size and volume underestimation by US was tumor-size-dependent, increasing with pathology-measured tumor size [37].

ILC does not have a specific appearance on US [38], most commonly presenting as a hypoechoic mass. Similar to mammography, ILC rarely presents as a well-circumscribed mass on US.

US plays a pivotal role in guiding biopsies of suspicious breast masses and detection of lymph nodes metastasis [39], although it still identifies less than 50% of lymph node metastasis [40]. US may play a role in predicting nodal burden in patients with ILC. Increased cortical thickness and loss of fatty hilum on US have been found to be associated with a higher nodal burden in patients with ILC (OR 58.40, 95% CI 5.09–669.71; *p* = 0.001) [30].

Overall, US remains an important imaging adjunct to mammography but likely provides no additional benefit for patients with newly diagnosed ILC given its underestimation of tumor extent and high discordance rate with final pathology.

### 3.3. Magnetic Resonance Imaging

Compared to other imaging modalities, breast magnetic resonance imaging (MRI) has the highest cancer detection rate; however, it is typically reserved for screening women at high risk for breast cancer due to cost, accessibility constraints, and false positivity rate [41]. In average-risk women with dense breasts, emerging evidence demonstrates significantly improved cancer detection rates with screening breast MRI compared to DBT, though there is currently insufficient evidence for widespread use in clinical practice [42].

For patients with newly diagnosed breast cancer, current guidelines recommend a preoperative breast MRI for the following groups: to identify occult primary breast cancer in patients with axillary node metastases and a negative mammogram, to better define tumor extent indeterminant by clinical exam and mammogram, to assess for potential fascia or muscle involvement in patients with invasive breast cancers contiguous with the chest wall, and to screen the contralateral breast in patients with a high risk of breast cancer and patients who are interested in reconstructive surgery after initial resection [43,44,45]. Breast MRI is also traditionally used to determine eligibility and tumor response to neoadjuvant endocrine (NAE) and chemotherapy (NAC) treatments [46].

Recently, breast MRI has been recognized as a useful diagnostic tool in patients with ILC due to various factors. For one, MRI has a much higher sensitivity rate for ILC tumor detection when compared to the other breast imaging modalities (mammography and ultrasonography). The sensitivity rates for MRI range from 83% to 100%, with most studies reporting a sensitivity rate greater than 95% [29,30,35,36,47,48,49]. Furthermore, recent advancements in MRI spatial resolution (e.g., 3T vs. 1.5T) are leading to increased detection of tumors that were previously occult on mammography and ultrasonography [50]. Fewer studies have investigated the specificity rates for MRI but estimates range from 87% to 92.4% [35,49].

An additional benefit of MRI is that breast density does not limit cancer detection rates. In fact, a group of researchers found that supplemental screening with MRI rather than mammography alone in women with very dense breast tissue enhanced detection and reduced the number of interval cancers during the screening period [51].

The literature consistently reports MRI as the imaging modality with the strongest correlation of imaging to pathology tumor size estimate for ILC. Studies report the correlation coefficient for MRI–pathology tumor size as 0.58 to 0.97 with the average mean difference ranging from 1.6 mm to 7 mm [31,33,34,36,47,48,49,52,53,54].

Although MRI appears to be the superior imaging modality for patients with ILC, there is still debate about the accuracy of MRI to determine maximum ILC tumor span [47]. Estimates for tumor size overestimation range from 26% to 36.7% [9,35,48,55,56], while estimates for underestimation range from 13.3% to 59.1% [9,33,34]. Regardless of the degree of under- or overestimation, most studies have found that MRI is consistently more accurate in estimating maximum ILC tumor extent when compared to conventional imaging (mammography and ultrasonography) [52]. Gest et al. performed a logistic regression analysis to investigate whether there are certain factors that may predict discordance between tumor size estimation by MRI and final pathology [57]. Menopausal status (OR 0.27, 95% CI 0.10–0.71; *p*  =  0.01), hormone receptor (HR) status (HR negative, OR 1.64, 95% CI 0.27–9.89; HR positive, OR 0.64, 95% CI 0.21–1.88; *p*  =  0.09), and NAC (OR 10.33, 95% CI 3.58–29.8; *p*  <  0.001) were all independently associated with greater overestimation of tumor size by MRI, while histological size (OR 1.05, 95% CI 1.02–1.08; *p*  <  0.0001) and the presence of an additional in situ component (OR 4.66, 95% CI 1.01–21.5; *p*  =  0.02) were associated with greater underestimation of tumor size by MRI [57]

One of the main benefits of including MRI as part of the diagnostic workup for patients newly diagnosed with ILC is the detection of additional ipsilateral and/or contralateral breast cancer. The data show that MRI detects an additional lesion in about one-third of patients with ILC and around 65–88% of these additional lesions are confirmed by pathology to be malignant [31,36,52,54,58,59,60,61]. One study found that 7% of patients with newly diagnosed ILC who underwent bilateral breast MRI benefited from early detection of contralateral breast cancer. A multivariate analysis by Wong et al. found that higher breast density (odds ratio 3.19; 95% CI 1.01 to 10.0) and lymph node positive disease (odds ratio 4.02; 95% CI 0.96 to 16.9) are significantly associated with additional suspicious findings on bilateral breast MRI in patients with ILC [29]. In patients with multifocal/multicentric breast cancer, MRI shows a high sensitivity for the detection of additional cancer foci with sensitivity estimates ranging from 88% to 91.17% [62,63]. Because ILC is more likely to be multifocal, multicentric, and bilateral than other types of breast cancer, patients with newly diagnosed ILC may greatly benefit from a bilateral breast MRI. In addition to earlier detection of additional ipsilateral and/or contralateral malignancies, MRI can also detect potential infiltration of the underlying pectoralis fascia or muscle layer that may not be visible on conventional imaging [31].

Due to the frequent detection of additional lesions, about 25–50% [9,29,35,36,49,52,54,58,64,65] of patients with ILC who undergo a bilateral breast MRI report a change in surgical management from breast-conserving surgery (BCS) towards total mastectomy. The most common changes in surgical management are more extensive unilateral surgery (wider excision or total mastectomy) or the addition of contralateral surgery. One study including patients with all types of breast cancer found that patients with ILC were the most likely to have their surgical management changed by preoperative breast MRI [66].

Another benefit of patients with ILC undergoing a preoperative MRI is the potential for improved prediction of nodal disease burden. MRI has been shown to be comparable to axillary US in the evaluation of axillary nodal status of these patients and an additional axillary US may only be required when there are suspicious nodal findings on MRI [67].

The prediction of nodal status is particularly important for patients with ILC after neoadjuvant therapy (NAC or NAE) as the upgrading or downgrading of axillary lymph nodes influences axillary surgery decisions. A study by Abel et al. found that a post-treatment/preoperative MRI is poor at predicting nodal status in patients with ILC after neoadjuvant therapy [68]. Sensitivity and positive predictive value were significantly higher in clinically node positive patients compared to clinically node negative patients (47.4% vs. 20%, *p*  =  0.0485 and 81.2% vs. 40%, *p*  =  0.0044) and specificity and negative predictive value were significantly higher in clinically node negative patients compared to clinically node positive patients (78.6% vs. 33.3%, *p*  =  0.0019 and 57.9% vs. 9.1%, *p*  =  0.0005) [68]. Additionally, for clinically node positive patients only, an abnormal post-treatment/preoperative MRI was associated with a significantly higher proportion of patients with high burden of nodal disease on pathology compared to patients with a normal post-treatment/preoperative MRI (61.1% versus 16.7%, *p*  =  0.034) [68]. These results imply that a preoperative MRI may be able to help predict high nodal burden and assist in axillary surgery planning in clinically node positive patients with ILC after neoadjuvant therapy. However, neoadjuvant therapy has been shown to not provide much benefit for patients with ILC tumors especially if clinically node negative [69]. Neoadjuvant therapy is associated with lower rates of downstaging and higher rates of margin positivity in cases changed from total mastectomy to BCS after neoadjuvant therapy [69]. Additionally, neoadjuvant therapy may not improve BCS rate and may lead to higher rates of total mastectomy after initial BCS [34,69]. However, for patients who are candidates for neoadjuvant therapy, studies show that MRI is useful in assessing tumor response after therapy [70]. As a result, for patients with ILC who undergo neoadjuvant therapy, MRI may play a role in assessing tumor response rate and staging after therapy to optimize surgical planning [70].

There continues to be debate in the literature about whether including MRI as part of the diagnostic workup for patients with ILC leads to unnecessary biopsies and mastectomies. In one study by Amin et al., MRI was found to have provided value by identifying additional malignancies and to have caused harm by leading to unnecessary additional biopsies almost equally in the study population of patients with ILC [71]. Rates of unnecessary biopsies due to preoperative MRI in patients with ILC range from 28% to 33.3% [14,71]. Additionally, many researchers have found that there are no unnecessary primary or final mastectomies as a result of patients with ILC undergoing a preoperative MRI [31,52,60,72,73]. Interestingly, a large cohort study by Lobbes et al. found that the likelihood of primary mastectomy as the chosen surgical treatment plan following preoperative breast MRI varied by breast cancer type. Patients with ILC who underwent a preoperative MRI had a lower likelihood of primary mastectomy compared to those who did not undergo an MRI (OR 0.86, 95% CI 0.76–0.99), while patients with IDC who underwent a preoperative MRI had a higher likelihood of primary mastectomy (OR 1.30, 95% CI 1.22–1.39) [74]. Some studies have also found a trend towards lower final mastectomy rates and secondary surgery rates in patients with ILC who undergo a preoperative MRI compared to patients who do not [72,75]. This trend is likely a result of the lower re-excision rates after initial BCS seen in patients with ILC who receive a preoperative MRI [48,54,60,75,76]. A study by Mann et al. found that 27% of patients with ILC in the non-MRI group compared to only 9% of patients with ILC in the MRI group required a re-excision surgery after initial BCS (OR 3.64, 95% CI 1.30–10.20; *p* = 0.010) [75]. However, many other studies have concluded that there is no difference in re-excision rates between MRI and non-MRI groups in patients with ILC [34,52,65,77,78]. In summary, the addition of preoperative MRI in the diagnostic workup of patients with newly diagnosed ILC is most likely associated with lower reoperation rates and minimal to no increase in the rate of initial or final mastectomies.

Although MRI has a higher sensitivity rate for ILC tumor detection, there has been concern about higher false positive rates compared to conventional imaging. In one study by Stivalet et al., researchers found that MRI sensitivity for ILC tumor detection was 100% at the expense of a 26% false positive rate [30]. The conclusions reached by this study, however, may be limited by its small sample size of only 15 patients with ILC and 31 total ILC masses. Several other studies have found that MRI does not in fact lead to a higher false positive rate in patients with ILC despite its high sensitivity rate [31,79,80]. Ultimately, it is recommended that all new lesions found on MRI should be evaluated by a US-guided or MRI-guided core biopsy to ensure pathologic proof of additional malignancy before a decision is made to change surgical management from BCS to ipsilateral, contralateral, or bilateral mastectomy [14,60]. This “second-look” biopsy by US or MRI can help to mitigate the high false positive rate with MRI and ensure a low rate of unnecessary operations [14,16,60].

In conclusion, some researchers support the addition of MRI to the diagnostic workup for all patients with ILC, especially when there is ambiguity of findings on clinical and conventional imaging (mammography and ultrasonography) [61,70]. However, the research shows that the benefits of adding MRI to the diagnostic workup for patients with ILC are limited to patients who are younger, have dense breasts, poor visualization by mammography, are post-neoadjuvant therapy, and/or are considering BCS [65]. Many societies have yet to determine whether MRI should or should not be included in the preoperative management of patients with ILC. The European Society of Breast Imaging and EUSOMA working group and NCCN recommend the use of preoperative MRI for patients with ILC [35,60]. For now, the addition of MRI to the diagnostic workup of patients with newly diagnosed ILC remains a case-by-case and institution-by-institution basis.

### 3.4. Contrast-Enhanced Mammography

Contrast-enhanced mammography (CEM), also referred to as contrast-enhanced spectral mammography (CESM) and contrast-enhanced digital mammography (CEDM), was approved by the Food and Drug administration in 2011. CEM combines conventional mammography with administration of intravenous iodinated contrast to provide both morphological and functional information about breast tissue and possible lesions. Commercially available CEM uses a dual-energy technique to generate two images in each projection: a low-energy mammogram similar to a standard mammographic image and a recombined image, highlighting areas of contrast enhancement while suppressing background glandular tissue [81]. Breast Imaging Reporting and Data System lexicon for CEM was recently introduced through the American College of Radiology to standardize interpretation and reporting [82].

In general, CEM is reported to have comparable accuracy to breast MRI, with equal to slight lower sensitivity and similar to slightly higher specificity [83,84,85,86]. In a recent meta-analysis, CEM had an overall 95% sensitivity and 81% specificity for breast cancer detection [87]. Additionally, CEM maintains favorable test characteristics in women with dense breasts, demonstrating a pooled 95% sensitivity and pooled 78% specificity in this cohort [87]. With regard to ILC, small single-center studies report CEM sensitivity ranges of 97–100% [83,88,89].

ILC more commonly presents on CEM as a mass rather than non-mass enhancement [88]. Cancers identified on CEM correlate well with histologic size, significantly outperforming mammography [90]. When comparing tumor size on CEM vs. MRI relative pathologic sizing, some studies show CEM having a superior correlation (Pearson correlation coefficient 0.75 CEM vs. 0.65 MRI, *p* < 0.01) [91] whereas in other studies MRI is superior (0.84 MRI vs. 0.77 for CEM, *p* < 0.01) [92]. In a study of 31 patients with ILC undergoing CEM, masses and non-mass enhancement had similar intraclass correlation coefficients for histologic sizing, 0.851 and 0.819, respectively [89].

The sensitivity of CEM in identifying multifocal, multicentric disease in the preoperative setting has been reported around 97.36–100% [88,89]. CEM has been reported to identify contralateral cancer in 10% [93]. For preoperative assessment of disease extent, CEM has been found to change surgical management in up to 20% of patients, including more extensive surgery in 16% (*n* = 16%) and conversion to mastectomy in 4% (*n* = 4) [83]. Given the accuracy of CEM in identifying the extent of disease, the surgical re-excision rate has been reported as low as 6.7% [89]. CEM has been reported to lead to additional biopsies in 12–19% of patients undergoing preoperative evaluation [83,93], though of the additional biopsies, up to 67% proved to be additional foci of invasive carcinoma [83].

The rates of change in surgical procedure and contralateral cancer detection on preoperative CEM are similar to MRI [59,66,71]. Preliminary data suggest that CEM may be equivalent to MRI in patients with breast cancer and a history of breast augmentation [94]. However, CEM does not allow for axillary staging and cannot visualize chest wall or internal mammary lymph nodes [81,95].

For women with breast cancer receiving NAC, MRI has traditionally been the main imaging modality used for assessing tumor response to NAC [96]. Conventional mammography has been of limited use and is not recommended to examine tumor response to NAC. The American College of Radiology grades the various imaging modalities for post-NAC treatment as follows: grade 9 for MRI, grade 8 for US, and grade 7 for mammography [96]. However, with the advent of CEM, which can reveal atypical vascular proliferation in tumors and increase the sensitivity of conventional mammography, studies have now found CEM to be comparable to MRI in determining the efficacy of NAC in breast cancer patients [96,97,98,99]. Some researchers advocate for the use of CEM as an alternative to MRI due to its shorter examination time, greater availability, better patient tolerance, and fewer contraindications [96,99].

Additional advantages of CEM relative to MRI include lower cost [100] and increased patient preference [101]. Exam time and interpretation time are faster with CEM, and there are no contraindications to metallic implants, patient size, or claustrophobia [102]. Disadvantages of CEM include relatively higher risks of contrast reactions [103,104] and potential contrast nephropathy [105] as compared to MRI, although the overall risks remain low. CEM also uses ionizing radiation [106], and CEM-guided biopsy is not yet widely available [107].

Although not specifically focusing on ILC, there are new clinical trials on the horizon investigating the utility of CEM in breast cancer screening and preoperative assessment. An upcoming prospective observational cohort study, Comparison of Breast Cancer Screening with CESM to DBT in Women with Dense Breasts (CMIST), plans to determine if CEM is more accurate than combined DBT and whole-breast US for primary screening in women with dense breasts [108]. Preoperative Contrast Enhanced Mammography in Staging of Malignant Breast Lesions (PROCEM trial) is an ongoing prospective randomized trial to evaluate the added value of CEM in the preoperative staging of breast malignancies [109]. Patients with breast cancer whose primary treatment is surgery will be randomized to no further imaging or CEM. The feasibility study prior to recruitment of this trial found that the treatment plan was changed in 10/47 cases (21%) and that CEM demonstrated improved size estimation with final pathologic size compared to mammogram and ultrasound [110].

### 3.5. Additional/Emerging Imaging Modalities

#### 3.5.1. Computed Tomography

The use of computed tomography (CT) in the detection of ILC has also been studied. Hikino et al. studied contrast-enhanced chest CT (CECT) to help determine and describe the pattern of morphologic and contrast enhancement features of ILC [111]. The features identified using CECT were characterized and divided into five groups: an ill-defined and inhomogeneous mass with or without regional heterogeneous enhancement, a spiculated inhomogeneous mass, a regional heterogeneous enhancement, and a normal finding. ILC has been reported to be non-localized with weak enhancement and low rate of enhancement on CT [112]. Many patients with ILC who were evaluated with CECT had tumors with gradual and late enhancement, which is typically seen in mastopathy and in other benign tumors, which may make evaluating the extent of disease and size of tumor for operative planning challenging with CECT. Despite this challenge, however, assessing extent of disease was more accurate for CECT compared to either mammogram or ultrasound (79% vs. 71%) [112]. Similarly, CT performed better than mammogram and ultrasound to determine the extent of disease after NAC (71% for CT, 53% for MMG, 48% for US). CECT performed better than MMG or US in the specific setting of determining extent of disease of ILC after NAC. On the contrary, in another retrospective review, no statistically significant difference was shown in the diagnostic rate of lesions in either the presence or suspicion of malignancy among mammography, ultrasound, or CECT [111]. All three modalities detected the ILC 94.4% of the time. The correct diagnosis of a malignant lesion was made 88.9% of the time for both mammography and CECT, and 72.2% of cases for ultrasound.

Pathologic tumor size has shown to be linearly related to size estimates on CECT. In one study, findings correlated with a Spearman’s rho correlation coefficient of 0.84 (*p* < 0.001) with 80.0% of tumors being concordant between the chest CECT and pathology [113]. However, these tumors included both IDC and non-IDC tumors, including ILC. Concordance was shown to be higher for IDC than non-IDC tumors (82.4% vs. 65.0% for IDC vs. non-IDC, *p* = 0.011), which is consistent with the known literature. In the evaluation of only ILC lesions in a smaller study, a Pearson correlation coefficient of r = 0.907, *p* < 0.01 was found, which outperformed mammogram and US (r = 0.698, *p* < 0.01 and r = 0.347, *p* = 0.19 for MMG and US, respectively) [111]. CECT was also shown to perform better than both mammogram and US combined (r = 0.705, *p* < 0.01).

Cone-beam breast computed tomography (CBBCT) is an emerging imaging technique whereby the breast is imaged while the patient is prone and the breast extends below the level of the table through an opening and allows for full 3D imaging [114]. Both non-contrast (NC-CBBCT) and contrast-enhanced (CE-CBBCT) studies were found to be superior to mammogram and ultrasound for identification of breast masses, particularly in women with dense breasts [114,115,116,117].

The sensitivity of CE-CBBCT was found to be higher compared to NC-CBBCT [114,115] and inferior compared to MRI [115]. The specificity of CE-CBBCT, on the other hand, was slightly lower than for MMG and NC-CBBCT; however, specificity of CE-CBBCT trended higher compared to MRI, though this was not statistically significant [115]. In one meta-analysis of six studies, the pooled sensitivity for NC-CBBCT was 0.789 (95% CI: 0.66–0.89) and pooled specificity was 0.697 (95% CI: 0.471–0.851) whereas the pooled sensitivity for CE-CBBCT was 0.899 (95% CI: 0.785–0.956) and pooled specificity was 0.788 (95% CI: 0.709–0.85) [116]. In this same meta-analysis, the diagnostic accuracy of CE-CBBCT was comparable to MRI [116].

As an imaging study, it has been found to be more comfortable than MMG given that it does not require compression of the breast, has a quick image acquisition, and is an option for patients with contraindications to MRI [114,116]. It also maintains its integrity for the evaluation of patients with implants [117]. CBBCT can also be used as the imaging modality for vacuum-assisted imaging-guided biopsies of suspicious lesions [114,116]. One of the primary limitations of CE-CBBBCT is the higher exposure of radiation due to the dual acquisition of images [117]. At present, there appears to be a paucity of data evaluating the use of CBBCT in the detection of ILC-specific malignancies.

#### 3.5.2. Molecular Breast Imaging/Breast Specific Gamma Imaging (technitium-99m-sestamibi)

Molecular breast imaging (MBI), also known as Breast Specific Gamma Imaging (BSGI), is a nuclear-medicine-based breast imaging modality that takes advantage of the fact that technitium-99m-stestamibi is taken up by breast cancer. Improvements in the technology have allowed MBI to deliver radiation doses similar to mammography [20,118]. The sensitivity of MBI for breast cancers has been shown to be 89–96.4% [118,119]. In one study, MBI demonstrated twice the cancer detection rate as compared to mammography [120]. Another study comparing the use of MBI vs. CEM and MRI for determining extent of disease found that MBI and CEM were effective for preoperative staging of breast cancers with increased specificity compared with MRI [121]. Unlike MRI, however, MBI is limited in its evaluation of the axilla and chest wall [121]. As a supplemental screening tool, MBI is inferior to MRI for detection of cancers [122,123].

In a retrospective study, the sensitivity of MBI for the detection of ILC has been reported as 93%, in comparison to mammography, sonography, and MRI with sensitivities of 79%, 68%, and 83%, respectively [124]. There is emerging literature that this imaging modality may have potential as a screening and diagnostic tool. As a functional study, MBI has improved cancer detection rates relative to women with dense breasts undergoing screening mammography [119,125]. Preliminary data from an ongoing trial demonstrate similar findings for women undergoing DBT vs. MBI [126,127]. This quality makes MBI a promising potential adjunct for the diagnosis and evaluation of ILC in patients with dense breasts [20,120]. ILC, however, has less sestamibi uptake compared to ductal carcinoma; as such, this results in lower detection rates by MBI, limiting its use at this time for preoperative evaluation in patients who can otherwise undergo MRI [128].

#### 3.5.3. Positron Emission Mammography & Positron Emission Tomography

Breast positron emission tomography (PET), also known as positron emission mammography (PEM), is a nuclear imaging technique that enhances focus on the breast rather than the whole body. It is not used for routine screening, but it can be used in place of MRI to determine the extent of breast disease in patients with newly diagnosed breast cancer [129]. A study by Narayanan et al. found that irregular or lobulated morphology and a BI-RADS category 3 assessment or greater on PEM were strong predictors of malignancy and likely warrant biopsy rather than follow-up [130]. Research shows that PEM imaging has greater sensitivity and diagnostic accuracy than whole-body PET/CT imaging in the detection of breast tumors [131,132]. However, PEM requires the PET camera to be configured like a mammogram machine and is not widely available. Additionally, although MRI and PEM may have comparable breast-level sensitivity rates, MRI tends to outperform PEM when it comes to lesion-level sensitivity, prediction of need for mastectomy, and positive predictive value of imaging-prompted biopsies [129,133].

At present, whole-body PET/CT is not routinely recommended for the staging of breast cancer. In one study evaluating the effectiveness of 18F-FDG PET/CT and MRI in ILC and IDC, the detection rate for ILC was 75% with a sensitivity and specificity for ipsilateral lesions of 0% and 91.7% [134]. Sensitivity for 18F-FDG PET/CT was significantly lower than for MRI (0% vs. 87.5%, *p* = 0.001); however, specificity was higher (91.7% vs. 58.3%, *p* = 0.008). 18F-FDG PET/CT also had fewer false positive cases for multiple contralateral lesions compared to MRI, though none of the false positives on 18F-FDG PET/CT were for ILC. Overall, tumor size and SUVmax did not correlate as well in the ILC group compared to the IDC group (Spearman correlation coefficient 0.25 (*p* = 0.179) for ILC vs. 0.57 (*p* < 0.001) for IDC). This study also confirmed prior studies that demonstrated that 18F-FDG PET/CT cannot accurately detect very small lymph node metastases and micrometastases and, therefore, cannot replace current guidelines for axillary staging via sentinel node biopsy. However, 18F-FDG PET/CT demonstrated a higher sensitivity for detection of axillary lymph node metastases than MRI in the ILC group, which is not consistent with prior studies and may be related to the small sample size considering the low sensitivity for primary and additional ipsilateral ILC lesions.

A prospective clinical trial was conducted to evaluate 18F-fluciclovine PET/CT for assessing the extent of invasive carcinoma of the breast as an alternative to 18F-FDG PET/CT [135]. In this study, all locally advanced breast cancers were 18F-fluciclovine–avid. Of the 21 patients with pathologically proven axillary nodal metastases (16 of 19 with IDC and 5 of the 8 with ILC), 18F-fluciclovine–avid axillary nodes were seen in 20. There were no false positive results. Concordance for SUVmax was weak (concordance correlation coefficient 0.04; CI −0.16–0.24). Concordance for metabolic tumor volume between 18F-fluciclovine and 18F-FDG was strong (concordance correlation coefficient 0.89; 95% confidence interval 0.73–0.96).

A recent pilot study by Eshet et al. investigated the role of 68Ga-fibroblast activation protein inhibitor (FAPI) PET/CT in the detection of non-18FDG PET/CT avid tumors in patients with known metastatic ILC [136]. Results were promising with significantly higher number of metastatic lesions identified by FAPI PET/CT compared to CT alone (*p* = 0.022). More research is warranted to determine the potential role for 68Ga-FAPI PET/CT in the systemic staging for patients with metastatic ILC.

#### 3.5.4. Radiomics/Artificial Intelligence

The use of artificial intelligence (AI) in the field of breast imaging has been ongoing since 1998 [137,138]. Computer-aided detection and diagnosis (CAD) models have been developed, and continue to be developed, as a means of assisting in the detection and diagnosis of breast cancer by optimizing workflow through accurate and efficient image interpretation [138,139]. Subsequent studies have demonstrated, however, that CAD models do not improve radiologist accuracy [140], though the goal is to reduce errors [139]. As such, efforts have been made to explore ways in which AI can assist in either or both accuracy and efficiency, which is where deep learning may play a role. As summarized by Madani et al., “AI models can generally be categorized into two groups to interpret and extract information from image data: (1) Traditional machine learning algorithms which need to receive handcrafted features derived from raw image data as preprocessing steps, (2) Deep learning algorithms that process raw images and try to extract features by mathematical optimization and multiple-level abstractions” [139].

This field includes machine and deep learning modalities that utilize convolutional neural networks (CNN), which can learn from data and make predictions [137]. Data have shown that CNNs have improved detection accuracy compared to traditional CAD for breast imaging and can also lead to reduced time to assess mammograms [137]. Deep learning methods have demonstrated that training with even relatively small datasets can lead to well-performing models [139]. Machine learning is being used to analyze mammogram results in combination with patient risk factors in order to calculate a more accurate probability of malignancy, and in doing so, improve the positive predictive value of recommendations for biopsy [141]. These systems have been applied to MMG, US, MRI, PET/CT, and histopathology [139].

One way in which AI is being used to assist is by deprioritizing likely negative mammograms so that radiologists can focus more time on the interpretation of abnormal studies [137]. It has been used in mammography to analyze images for detection and classification of breast masses and microcalcifications, as well as assess breast density, assess breast cancer risk, and improve image quality [141]. Another application of these technologies is to be able to identify the hormone receptor status or genetics of a cancer by imaging alone as well as predict tumor staging and/or response to therapies [138].

Radiomics is a technology that uses a computation of quantitative imaging features of an abnormality or of tumor features in order to predict clinical outcomes, which can then be analyzed with machine learning to continue improving [138,142]. Radiomics is currently being applied to MMG, DBT, US, MRI, and PET/CT [138,143]. One study compared radiomics analysis of CEM and dynamic contrast-enhanced MRI [143]. In this study, radiomics analyses resulted in high accuracies for both imaging modalities when classifying findings as invasive or non-invasive, as well as hormone receptor status, and tumor grade [143]. Davey et al. suggest that, with data from systematic reviews and meta-analyses, radiomics has comparable diagnostic accuracy to core needle biopsy for differentiating breast cancer molecular subtypes and may be a practical alternative [142].

Conclusions are limited regarding the use of radiomic evaluations for ILC given its low prevalence, and hence low representation in study samples. It is proposed that the low density and modest detection rates of ILC on mammography may also limit the use of radiomics for this tumor subtype [142]. The field of radiomics and AI is still relatively new, and it is uncertain whether it may play a role in the diagnosis or preoperative evaluation of ILC.

## 4. Discussion

Our review of the literature finds that MRI and CEM clearly surpass US, DM, and DBT in terms of sensitivity, specificity, ipsilateral and contralateral detection, concordance, and estimation of tumor size and extent of disease within the breast for ILC. Importantly, both MRI and CEM have each been shown to enhance surgical outcomes in patients with newly diagnosed ILC that had one of these imaging modalities added to their preoperative workup.

Recent studies have identified some of the key features of ILC on MRI and CEM, the two imaging modalities with the highest ILC detection and size correlation rates, in the hopes of better characterizing its unusual presentation. For MRI, the absence of smooth margins and rim-shaped enhancement are the key features associated with ILC [30]. Although ILC presentation is highly variable, four major patterns have been identified on MRI with most ILC presenting as a single spiculated non-homogenous mass or a dominant lesion surrounded by many small enhancing foci [70,144]. Similar patterns are described in CEM with lesions presenting as a focus, mass, non-mass enhancement, and a mass with closely associated non-mass enhancement [89]. For both imaging modalities, many ILC tumors only show weak enhancement when compared to IDC. Despite MRI and CEM showing promise for increased ILC detection, ILC continues to remain a diagnostic challenge due to its typically weak enhancement, linear growth pattern, and lack of a desmoplastic reaction, calcification, necrosis, or hemorrhage.

Despite the potential benefit of additional imaging studies for the staging of ILC, MRI and CEM are not recommended as part of breast cancer screening guidelines. Screening mammography remains the only imaging modality that has been proven to reduce mortality of breast cancer. MRI only has evidence of a screening benefit in patients who are known BRCA mutation carriers and their first-degree relatives, as well as patients who have a calculated lifetime risk of breast cancer > 20–25% by a model dependent largely on family history [145]. This review does not recommend the addition of MRI or CEM to screening guidelines and only sets out to make recommendations regarding the preoperative staging and preoperative evaluation for ILC based on current data.

The performance of MRI and CEM regarding their role in the workup and preoperative evaluation of ILC is delineated in Table 1. These two imaging modalities outperform standard digital mammography, tomosynthesis, and ultrasound in accurately assessing tumor size and extent of disease, including identification of multifocal/multicentric and contralateral disease, in patients with ILC. Given their increased concordance for ILC lesions, MRI and CEM have been shown to improve surgical outcomes in patients where one of these imaging modalities was included in the preoperative evaluation.

We recommend the addition of a preoperative MRI or CEM for more accurate estimation of tumor size and extent of disease given the propensity for ILC to be multifocal, multicentric, bilateral, and/or poorly detected by conventional imaging (MMG and US). While both MRI and CEM have similar performance, there are advantages to each study, which are summarized in Table 2. Despite the limited clinical benefit of NAC or NAE for patients with ILC, we also support the use of either MRI or CEM to assess tumor response for the small subgroup of patients undergoing NAC or NAE. MRI and CEM appear to have similar performance for the accurate estimation of tumor size and extent, and we recommend either for surgical planning, particularly for patients planning to pursue BCS.

There currently remain no established indications for the use of CEM as part of the diagnostic workup for patients with any breast cancer subtype. However, due to CEM’s comparable diagnostic accuracy to MRI, it may be considered when MRI is not available or feasible. CEM also has the benefit of a shorter examination with shorter image acquisition and interpretation time. It is also well-tolerated by patients who may be unable to undergo MRI due to claustrophobia or issues lying flat for extended periods of time. This imaging modality is estimated to be a relatively lower cost investment compared to MRI in that most mammography systems can be retrofitted to perform CEM. One downside to CEM is its inability to assess the entire axillary region. However, the addition of an axillary US with CEM helps to address this barrier. In contrast, MRI and CT have been shown to have superior axillary region visualization. Therefore, we recommend the use of MRI or CEM with axillary US as part of the initial workup of patients with newly diagnosed ILC to better predict the extent of nodal disease. Additionally, we recommend the use of MRI to help guide axillary surgery planning for clinically node positive patients with ILC after NAC.

Locoregional staging and determining the extent of disease are important for surgical planning to plan an adequate resection. This is because both multifocal and multicentric disease for ILC have similar prognosis to unifocal ILC lesions, provided a complete resection. At present, there is no evidence to support an overall reduction in recurrence- or disease-free survival with use of a preoperative MRI. Given this, in addition to the high rate of false positive findings on MRI, surgical plans should not be modified based on MRI findings alone in the absence of pathologic confirmation with biopsy. Furthermore, we recommended that a preoperative MRI not be recommended to patients at institutions where MRI-guided biopsies cannot be performed as MRI-only identified lesions can otherwise not be appropriately biopsied. Currently, the capability to perform CEM-guided biopsies is available in very few centers.

In a recent review, researchers concluded that the use of fast sequences or an abbreviated protocol may be an acceptable alternative to the full diagnostic protocol for breast MRI by reducing examination time and cost while maintaining diagnostic accuracy [146]. More research is warranted, but these findings suggest that the adoption of an ultrafast or abbreviated protocol for MRI may help to enhance breast cancer detection without the additional cost and time of full-protocol MRI.

Regarding additional and emerging imaging modalities for evaluation of patients with ILC, CT, PET/CT, and MBI have not been shown to perform as well as MRI or CEM based on the current literature. Proponents of CECT imaging suggest that CECT is better for surgical planning than MRI and CEM as the breast is imaged in the same supine position as the patient would be positioned surgically [113]. However, given the standard practice of localizing nonpalpable lesions prior to surgical excision and the experience of surgeons in utilizing current imaging modalities for surgical planning, the clinical significance of this is likely low. There may be promise for PET/CT for the identification of extraaxillary nodal disease, though it has not yet demonstrated benefit for identification of small axillary metastases and thus cannot replace sentinel node biopsy [134]. Given the fairly high false negative rates, PET/CT is still not recommended in the preoperative evaluation for patients with ILC. Finally, MBI has early data to suggest detection rates rivaling those of MRI and CEM. There are, however, limitations such as higher radiation dose to patients and longer acquisition times [20]. There is also not yet an optimal modality for obtaining imaging-guided biopsies for lesions that may only be seen on MBI [124].

## 5. Conclusions

Clinical examination along with conventional imaging by MMG, DBT, and/or US remains critical to the initial investigation of a new breast concern. However, these imaging modalities underperform in sensitivity, tumor size estimation, and detection of multifocal/multicentric disease for determining the extent of disease in cases of ILC. As a result, subsequent imaging is recommended for the preoperative evaluation of new ILC. Ultimately, the decision to pursue further diagnostic imaging with MRI or CEM should be a shared decision-making process between the patient and provider. Patients should understand the risks and benefits to additional imaging and how the imaging results may impact surgical management, risk of re-excision, and risk of recurrence. Of particular importance is that patients with newly diagnosed ILC interested in BCS are advised to undergo a preoperative MRI or CEM due to the additional information it can offer for surgical planning and management to improve rate of excision with negative margins and reduce reoperation rates.

## Figures and Tables

**Table 1 healthcare-11-00746-t001:** A comparison of the main variables and corresponding data that impact the diagnostic workup of ILC tumor(s) using MMG, US, MRI, and CEM (references noted within the text).

	MMG	US	MRI	CEM
Imaging–pathology size correlation	0.24–0.27	0.678–0.957	0.58–0.97	0.858–0.937
Sensitivity rate	34–92%	68–92%	83–100%, most >95%	97.36–100%
Multifocal/multicentric detection rate	24%	80.8%	89–91.17%	84.2–97.36%
Change in surgical planning (from BCS to more extensive unilateral surgery)			25–50%	13.3–20%
Rate of re-excision	6.7%		9%	6.7%
False positive detection rate	8–24%	22%	26–33.3%	19.3–33.3%

**Table 2 healthcare-11-00746-t002:** A comparison of the main advantages of MRI vs. CEM *.

	MRI	CEM
Results independent of breast density	+ + +	+ + +
Evaluation of the axilla	+ + +	+
Use in neoadjuvant setting	+ + +	+ + +
Biopsy capability	+ +	+ **
Compatible with implanted devices and retained metals	+	+ + +
Patient preference	+	+ +
Cost	+ + +	+
Ionizing radiation	n/a	+ + +
Risk of contrast reaction	+	+ +
Examination duration	+ +	+
Image interpretation time	+ +	+
Equipment availability	+	+ +

* The number of + signs indicates either increasing relative capability or amount in each category. ** CEM biopsy FDA approved, currently not widely available.

## Data Availability

Not applicable.

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
