# Peer review of "Invasive Lobular Carcinoma: A Review of Imaging Modalities with Special Focus on Pathology Concordance"

_healthcare, 2023, doi:10.3390/healthcare11050746_

Round 1

Reviewer 1 Report

An exhaustive work, which addresses an extremely current issue and which has seen an extremely spectacular but also rapid technological evolution at the same time. The work is classified as a narrative review. 

- In the search strategy, the years between which the bibliographic material was searched and identified (beginning year - final year) are missing. Also missing are some absolutely necessary terms in the search strategy, terms that, later in the material of the article, are presented for example Digital Breast Tomosynthesis, Ultrasonography, Computed Tomography, Positron Emission Mammography & Positron Emission Tomography. Also, the inclusion or exclusion criteria for some articles or various publications are missing.

- Some abbreviations are not explained in the text, for example DBT (line 80)

- The limits of mammography as an indication and as a sensitivity of the diagnostic result are not found – for ex high-density breast tissue and in young women in particular

- Bibliographic reference (line 100) 26 says explicitly - “Mammography is not widely available in all countries, and breast cancer incidence is increasing. We considered performance characteristics using ultrasound (US) instead of mammography to screen for breast cancer. Conclusions. Cancer detection rate with US is comparable with mammography''. The authors have another interpretation of the bibliographic material - "Whole-breast ultrasonography (US) is not widely used for breast cancer screening given the higher rate of false positives relative to mammography [26]". An adjustment of the authors' opinion seems necessary.

- Perhaps the most important argument for completing the diagnosis of the primary tumour with CEM and MRI is the one offered by the first step of the breast investigation, namely Mammography, Digital Breast Tomosynthesis, Ultrasonography. The doubts and ambiguities of these evaluations are a good reason for supplementing with CEM and MRI. Anyway, in the first stage, the patients arrive at the primary screening methods and only after that, gradually, the sensitivity of the diagnostic method increases. Among other things, there is no mention of the clinical examination, as the first diagnostic step, and the subsequent guidance to additional imaging methods.

- I did not identify in the text of the article what represents a complete diagnosis of the disease according to the TNM criteria. Only the identification of the primary tumour does not appear to be sufficient for diagnosis. Tumour extension - nodal metastases, distant metastases - should benefit from some comments by the authors on specific imaging investigations. An example, positron emission mammography (PEM) really has an important role in the diagnosis of primary breast tumours, but whole-body PET-CT is much more limited from this point of view. The role of whole-body PET-CT is precisely to complete the diagnosis, to allow an evaluation of the oncological treatment and to remotely investigate the therapeutic results. Regarding whole-body PET-CT (line 453), the statement by the authors "cannot accurately detect very small lymph node metastases and micro metastases" is somewhat strange. For this reason, can we exclude the diagnosis offered by whole-body PET-CT? Does MRI or CEM provide this information? Each method has its limits, the key to diagnosis and of course optimal therapy is the employment and gradual augmentation of specific investigations.

Author Response

An exhaustive work, which addresses an extremely current issue and which has seen an extremely spectacular but also rapid technological evolution at the same time. The work is classified as a narrative review. 

- In the search strategy, the years between which the bibliographic material was searched and identified (beginning year - final year) are missing. Also missing are some absolutely necessary terms in the search strategy, terms that, later in the material of the article, are presented for example Digital Breast Tomosynthesis, Ultrasonography, Computed Tomography, Positron Emission Mammography & Positron Emission Tomography. Also, the inclusion or exclusion criteria for some articles or various publications are missing.

              The methods section was revised to include years, and additional search terms that were accidentally omitted.

- Some abbreviations are not explained in the text, for example DBT (line 80)

Thank you for pointing out this oversight by the authors.  We have addressed this issue.

- The limits of mammography as an indication and as a sensitivity of the diagnostic result are not found – for ex high-density breast tissue and in young women in particular

              The limitations of mammography with regards the dense breast tissue were addressed in lines 76-82.

- Bibliographic reference (line 100) 26 says explicitly - “Mammography is not widely available in all countries, and breast cancer incidence is increasing. We considered performance characteristics using ultrasound (US) instead of mammography to screen for breast cancer. Conclusions. Cancer detection rate with US is comparable with mammography''. The authors

have another interpretation of the bibliographic material - "Whole-breast ultrasonography (US) is not widely used for breast cancer screening given the higher rate of false positives relative to mammography [26]". An adjustment of the authors' opinion seems necessary.

The study that is being referred found there were more false positives associated with US relative to mammography and recommended use of mammogram when available for screening. ‘…we found false positives more common in younger women on US but not mammography. In our study, with increasing breast density, false positives increased for US but not mammography…’.

- Perhaps the most important argument for completing the diagnosis of the primary tumour with CEM and MRI is the one offered by the first step of the breast investigation, namely Mammography, Digital Breast Tomosynthesis, Ultrasonography. The doubts and ambiguities of these evaluations are a good reason for supplementing with CEM and MRI. Anyway, in the first stage, the patients arrive at the primary screening methods and only after that, gradually, the sensitivity of the diagnostic method increases. Among other things, there is no mention of the clinical examination, as the first diagnostic step, and the subsequent guidance to additional imaging methods.

              The reviewer makes an excellent point in that it is the culmination of the investigations that increase the sensitivity, which supports our conclusion that additional studies are recommended in the preoperative setting after a diagnosis of ILC and that both MRI and CEM are similar in their performance to meet this need.  We added a comment regarding clinical examination both in the introduction and conclusion.

- I did not identify in the text of the article what represents a complete diagnosis of the disease according to the TNM criteria. Only the identification of the primary tumour does not appear to be sufficient for diagnosis. Tumour extension - nodal metastases, distant metastases - should benefit from some comments by the authors on specific imaging investigations. An example, positron emission mammography (PEM) really has an important role in the diagnosis of primary breast tumours, but whole-body PET-CT is much more limited from this point of view. The role of whole-body PET-CT is precisely to complete the diagnosis, to allow an evaluation of the oncological treatment and to remotely investigate the therapeutic results. Regarding whole-body PET-CT (line 453), the statement by the authors "cannot accurately detect very small lymph node metastases and micro metastases" is somewhat strange. For this reason, can we exclude the diagnosis offered by whole-body PET-CT? Does MRI or CEM provide this information? Each method has its limits, the key to diagnosis and of course optimal therapy is the employment and gradual augmentation of specific investigations.

              The focus of this paper was to mainly assess the imaging concordance of these studies with regards to the local/regional extent and comment on detection of metastatic disease

when applicable. It is beyond the scope of this review to comment on indications or value of whole body PET/CT(or other modalities) with regards to identification of metastatic disease. In line 465 it is noted that whole body PET/CT is not routinely recommended for the staging of breast cancer.  

Reviewer 2 Report

This review summarizes the conventional and newly emerging imaging modalities being used in the detection and determination of the extent of Invasive lobular cancer (ILC). The review also highlights the advantages of incorporating either MRI or CEM to the preoperative process of patients with newly diagnosed ILC, which could improve surgical outcomes. Overall, the authors have provided a comprehensively detailed insight into the pitfalls and benefits of various modalities, and very minor revisions are required before accepting the manuscript.

-Some abbreviations need to be cited in text when first used (MMG, DBT).

-The authors may also consider including a pilot trial suggesting a role of 68 Ga-FAPI PET/CT in ILC in their CT section (PMID: 36638243)

Author Response

This review summarizes the conventional and newly emerging imaging modalities being used in the detection and determination of the extent of Invasive lobular cancer (ILC). The review also highlights the advantages of incorporating either MRI or CEM to the preoperative process of patients with newly diagnosed ILC, which could improve surgical outcomes. Overall, the authors have provided a comprehensively detailed insight into the pitfalls and benefits of various modalities, and very minor revisions are required before accepting the manuscript.

-Some abbreviations need to be cited in text when first used (MMG, DBT).

Thank you for also bringing this oversight to our attention.  We have addressed this issue and used these abbreviations more consistently throughout.

-The authors may also consider including a pilot trial suggesting a role of 68 Ga-FAPI PET/CT in ILC in their CT section (PMID: 36638243)

Thank you for this suggestion.  While this pilot study did not specifically address the concordance of tumor size on 68Ga-FAPI PET/CT to final pathologic size, we have referred to this study in section 3.5.3 as it relates to the systemic evaluation of metastatic disease in ILC.  

Reviewer 3 Report

This is a comprehensive review; however, please see the below concerns:

1) The Methods section is extremely short:

·       How were the keywords selected?

·       Were synonyms considered?

·       Not clear what “various combinations” means.

·       Was the search performed manually or via an API?

·       Which fields (title, abstract etc.) were searched?

·       How were “Selected articles” filtered?

·       Are all the publications considered in English? Or the articles only?

Please expand this section. A figure showing an overview of the search process can be added here.

2) I recommend adding the numbers for all modalities to Table 1 for easier comparison.

3) The section between lines 228 and 252 shows a mix of trends. I suggest relaxing the conclusion here.

4) Please clarify the +/- scale in Table 2.

5) This paragraph is confusing:

“Invasive lobular cancer (ILC) is the second most common type of breast cancer after invasive ductal cancer (IDC) with an estimated 28,750 new cases in 2022[1]. Even though ILC comprises only about 10% of new breast cancer diagnoses, its incidence is similar to that of ovarian cancer (estimated 19,880 new cases in 2022) and two times of uterine cervix cancer (estimated 14,100 new cases in 2022)[1].”

Please add the location (United States). Instead of ‘similar’ and ‘two times of’, I suggest using ‘high, similar to’.

6) Please clarify or improve the wording for reported statistics if needed

e.g., does “…ILC tumor size by an average of 18%-53%...” refer to size or number of patients? “… in an average of n% of patients…”?

e.g., “…Mammogram remains the only imaging modality that has been proven to reduce mortality of breast cancer…” or mammography screening”?

Author Response

This is a comprehensive review; however, please see the below concerns:

1) The Methods section is extremely short:

  • How were the keywords selected?

Keywords were selected to be able to capture all imaging modalities that are used in the clinical setting.

  • Were synonyms considered?

Synonyms were not considered as the authors felt the search with selected keywords was successful in yielding relevant results.

  •  

 Not clear what “various combinations” means.

This was clarified in the manuscript.

  • Was the search performed manually or via an API?

This was addressed in the manuscript.

  • Which fields (title, abstract etc.) were searched?

This was addressed in the manuscript.

  • How were “Selected articles” filtered?

This was addressed in the manuscript.

  • Are all the publications considered in English? Or the articles only?

This was addressed in the manuscript.

Please expand this section. A figure showing an overview of the search process can be added here.

In addition; the methods section was revised to include years, and additional search terms that were accidentally omitted.

2) I recommend adding the numbers for all modalities to Table 1 for easier comparison.

              We have included mammography and ultrasonography to this table for ease of comparison for readers.

3) The section between lines 228 and 252 shows a mix of trends. I suggest relaxing the conclusion here.

              We agree with this assessment on review of the data and trends presented and have adjusted our stance to reflect this.

4) Please clarify the +/- scale in Table 2.

A footnote was added to clarify the scale and significance of the + units used.

5) This paragraph is confusing:

“Invasive lobular cancer (ILC) is the second most common type of breast cancer after invasive ductal cancer (IDC) with an estimated 28,750 new cases in 2022[1]. Even though ILC comprises only about 10% of new breast cancer diagnoses, its incidence is similar to that of ovarian cancer (estimated 19,880 new cases in 2022) and two times of uterine cervix cancer (estimated 14,100 new cases in 2022)[1].”

Please add the location (United States). Instead of ‘similar’ and ‘two times of’, I suggest using ‘high, similar to’.

              This paragraph has been updated for clarification.

6) Please clarify or improve the wording for reported statistics if needed

e.g., does “…ILC tumor size by an average of 18%-53%...” refer to size or number of patients? “… in an average of n% of patients…”?

              This statement was clarified within the text as noted in lines 118-119.

e.g., “…Mammogram remains the only imaging modality that has been proven to reduce mortality of breast cancer…” or mammography screening”?

              These statements have been clarified within the text as can be seen in lines 564-565.

Round 2

Reviewer 3 Report

Thank you, the manuscript has improved.